# Gut Microbiota Enterotype as a Predictor of Sarcopenia in the Japanese Elderly Population

**DOI:** 10.3390/nu17203250

**Published:** 2025-10-16

**Authors:** Sayaka Hotta, Michiko Matsunaga, Akimitsu Miyake, Aya K. Takeda, Satoshi Watanabe, Naoki Hosen, Keisuke Hagihara

**Affiliations:** 1Department of Respiratory Medicine and Clinical Immunology, Graduate School of Medicine, Osaka University, Suita 565-0871, Osaka, Japan; s_hotta@kanpou.med.osaka-u.ac.jp; 2Graduate School of Education, Kyoto University, Kyoto 606-8501, Japan; paprika3c5@gmail.com; 3School of Medicine, Tohoku University, Sendai 980-8575, Japan; akimitsu.miyake.d5@tohoku.ac.jp; 4Cykinso, Inc., Tokyo 151-0053, Japan; takeda@cykinso.co.jp (A.K.T.); s.watanabe@cykinso.co.jp (S.W.); 5Department of Hematology and Oncology, Graduate School of Medicine, Osaka University, Suita 565-0871, Osaka, Japan; hnaoki@bldon.med.osaka-u.ac.jp; 6Research Institute for Microbial Diseases (RIMD), Osaka University, Suita 565-0871, Osaka, Japan

**Keywords:** gut microbiota, enterotype, ET-B2, sarcopenia

## Abstract

**Background/Objectives**: Frailty and sarcopenia are age-related conditions that impair quality of life in older adults. Although the gut microbiota affects muscle health, its role in sarcopenia remains unclear. This study investigated the association between gut microbiota enterotypes and sarcopenia in community-dwelling older Japanese adults. **Methods**: In this cross-sectional study, 322 community-dwelling adults from the Japanese Frailty Scale cohort aged ≥65 years were assessed for sarcopenia using standardized criteria. Physical measures included grip strength, gait speed, and skeletal muscle mass index (SMI). Gut microbiota profiles were analyzed using 16S rRNA sequencing and classified into four enterotypes (ET-B1, ET-B2, ET-R, ET-P). Associations with sarcopenia were evaluated using multivariable logistic regression. **Results**: Participants with enterotype ET-B2 had significantly lower microbial diversity (*p* < 0.01) and reduced grip strength (*p* < 0.05), whereas the difference in SMI compared with ET-P individuals did not reach statistical significance (*p* = 0.0625). Sarcopenia prevalence differed significantly between enterotypes (*p* < 0.01). A predictive model incorporating age and ET-B2 exhibited an area under the receiver operating characteristic curve (AUC) of 0.795, significantly higher than the age-only model (AUC = 0.686, DeLong’s test, *p* < 0.05). **Conclusions**: Gut microbiota composition, especially enterotype ET-B2, is significantly associated with sarcopenia in older Japanese adults. These findings indicate the potential for using the gut microbiota as a biomarker and therapeutic target in treating age-related muscle decline.

## 1. Introduction

The rapid progression of population aging has become a global issue [1,2,3]. Elderly individuals are at a high risk of developing frailty, which leads to increased medical and long-term care costs, thereby placing a significant financial burden on social security systems [4,5]. Moreover, frailty has been shown to negatively impact clinical outcomes [6], making the development of preventive strategies an urgent priority.

Frailty is a multifactorial condition involving the musculoskeletal, immune, and endocrine systems, with immunosenescence and chronic inflammation playing central roles [7,8,9]. Recent studies have reported that gut microbiota is closely associated with metabolism [10,11], nutrition [12,13], and immunity [14]. Animal studies have demonstrated that aging-induced dysbiosis increases intestinal permeability, leading to systemic chronic inflammation [15]. Indeed, gut microbiome dysbiosis has been associated with a variety of diseases, such as inflammatory bowel disease (IBD) [16], cancer [17], cardiovascular diseases [18], and dementia [19]. Moreover, other studies have demonstrated a relationship between gut microbiota enterotype and statin therapy in obesity [20].

We previously developed a simple and promising screening test for frailty, the Japanese Frailty Scale (JFS) [21]. The JFS consists of five indicators (nocturia, low back pain, cold hypersensitivity, exhaustion, and age) and has been validated among community-dwelling older adults. In addition, the JFS showed strong relationship with the Kihon checklist (KCL), Locomo 5 and GDS-15 in the other assessment of frailty; however, the JFS exhibited low correlation with measures of physical performance such as grip strength and skeletal muscle mass index (SMI). Recent studies have reported that the gut microbiota may be associated with frailty and sarcopenia [22,23,24,25,26,27]. However, these studies have several limitations, such as small sample sizes and a lack of standardized diagnostic criteria for frailty or sarcopenia.

In the present study, we investigated the relationship among gut microbiota composition, sarcopenia and frailty in community-dwelling older adults using previously collected cohort data and metagenomic analysis. Based on our data, the gut microbiota profiles were classified into four enterotypes (ET-B1, ET-B2, ET-R, ET-P). The prevalence of sarcopenia differed significantly among the four enterotypes. A predictive model incorporating age and ET-B2 demonstrated significantly higher predictive performance than the model including age alone. In contrast, the genus-level analysis did not yield clear associations. Our findings identify a specific gut microbiota enterotype associated with sarcopenia, suggesting the potential of the gut microbiota as a biomarker for age-related muscle decline.

## 2. Materials and Methods

### 2.1. Study Population

Samples and data were collected from the development cohort of the JFS [21]. Participants in Study 1 (Cohort 1: cross-sectional study) were recruited in cultural circles organized by the town of Sango, Nara prefecture during May to June and October to December 2019, as described in previous study. The inclusion criteria were age ≥ 65 years and ability to participate in social activities in their town of residence. The exclusion criteria were individuals certified as needing long-term care and those with missing data for fecal microbiota analysis.

Participants in Study 2 (Cohort 2: longitudinal study) included those from Cohort 1 who underwent both a health check-up and fecal microbiota analysis twice. This included analysis of physical activity and body composition, with fecal microbiota analysis, twice during the period October to December 2019. Individuals with only a single fecal microbiota analysis were excluded. This study followed the TRIPOD (Transparent Reporting of a Multivariable Prediction Model for Individual Prognosis or Diagnosis) reporting guidelines [28].

### 2.2. Health Check-Up for Frailty and Related Conditions

On the day of the health check-up, age and sex were ascertained as demographic indicators, and height, body weight, hand grip strength, gait speed, two-step values, and body composition were measured as physical parameters, as the previously reported [21]. Data were collected by laboratory staff, including physicians, who followed a shared manual and a standardized data collection protocol.

Hand grip strength was measured twice on each hand in a standing position using a Smedley dynamometer (TKK5401, Takei Scientific Instruments Co., Ltd., Niigata, Japan). The higher value of the two was used for analysis.

Gait speed is considered an indicator of physical performance of older adults and a strong predictor of adverse outcomes [29]. To measure gait speed, participants were asked to walk 11 m at a normal pace. The 11 m section was divided into a 5 m section that was timed, with 3 m sections before and after to allow for acceleration and deceleration. To calculate two step value, the distance between two maximum stride steps divided by body height.

Body composition was determined using bioelectrical impedance analysis (InBody770; InBody, Seoul, Republic of Korea) to measure body mass index (BMI), phase angle, and SMI.

Japanese-Cardiovascular health study (J-CHS) criteria was proposed by the Japanese Geriatrics Society in 2015, as a modified form of the CHS criteria [30]. The KCL was comprehensive self-reported questionnaire, developed by the Japan’s Ministry of Health, Labour and Welfare, and widely used in Japan to identify older adults with frailty [31]. The KCL assesses domains such as instrumental of daily living, physical functioning, nutritional status, oral functioning, social activities, cognitive functioning, and depressed mood. The JFS is a frailty screening scale based on the aging concept of Kampo medicine, which evaluates five criteria: nocturia, low back pain, cold hypersensitivity, exhaustion, and age. Sarcopenia is partially overlapping with frailty, although it is focused on the musculoskeletal system and characterized by the loss of muscle mass and a reduction in muscle strength. Sarcopenia was assessed in this study based on criteria of the Asian Working Group for Sarcopenia-2019 [32].

### 2.3. Fecal Sampling, DNA Extraction, and Sequencing

Fecal samples were collected at home using brush-type collection kits containing guanidine thiocyanate solution (Techno Suruga Laboratory, Shizuoka, Japan), transported at ambient temperature, and stored at 4 °C. DNA was extracted from the fecal samples using an automated DNA extraction system (GENE PREP STAR PI-480, Kurabo Industries Ltd., Osaka, Japan) according to the manufacturer’s protocol. The V1–V2 region of the 16S rRNA gene was amplified using a forward primer (16S_27Fmod: TCG TCG GCA GCG TCA GAT GTG TAT AAG AGA CAG AGR GTT TGA TYM TGG CTC AG) and a reverse primer (16S_338R: GTC TCG TGG GCT CGG AGA TGT GTA TAA GAG ACA GTG CTG CCT CCC GTA GGA GT) and KAPA HiFi HotStart ReadyMix (Roche, Basel, Switzerland). For sequencing 16S amplicons using an Illumina MiSeq platform (Illumina, San Diego, CA, USA), dual index adapters were attached using a Nextera XT Index kit (Illumina). Each library was diluted to 5 ng/µL, and equal volumes of the libraries were mixed to 4 nM. The DNA concentration of the mixed libraries was quantified by qPCR using KAPA SYBR FAST qPCR Master Mix (KK4601, KAPA Biosystems [Roche] using primer 1 (AAT GAT ACG GCG ACC ACC) and primer 2 (CAA GCA GAA GAC GGC ATA CGA). The library preparations were carried out according to the Illumina 16S library preparation protocol. Libraries were sequenced using the MiSeq Reagent kit v2 (Illumina; 500 Cycles), to produce 250 bp paired-end reads.

### 2.4. Taxonomy Assignment Based on 16S rRNA Gene Sequencing

Data processing and assignment were conducted as follows using the QIIME2 pipeline (version 2020.8) [33] as follows: (1) joining paired-end reads, filtering, and denoising using a divisive amplicon denoising algorithm (DADA2) and (2) assigning taxonomic information to each amplicon sequence variant (ASV) using a naive Bayes classifier in the QIIME2 classifier. The classifier was trained using a robust taxonomy simplifier for SILVA (arts-SILVA), which was originally developed from the 16S rRNA taxonomy dataset based on SILVA 138 [34]. arts-SILVA was developed for the purpose of making Mykinso testing reports easier to understand for those who are unfamiliar with the complex rules of taxonomic nomenclature [35]. arts-SILVA simplifies the resulting taxonomic assignments by removing study-related labels, curating notable mis-entries, and generalizing uncommon names in the SILVA database. To obtain arts-SILVA, the V1V2 regions of the SILVA reference sequences were extracted and clustered according to the original manuscript for QIIME2 preparation in SILVA. Subsequently, some unnecessary/seemingly miss-labeled entries were deleted. (i.e., we removed labels with little information such as “D_6__unclutured bacteria”) and duplicate entries were collected; for example, “D_0__Bacteria;D_1__Bacteria Firmicutes” was corrected to “D_0__Bacteria;D_1__Firmicutes”). Subsequently, unnecessary taxa, such as “D_6__human metagenome”, were removed by manual inspection, and a consensus taxonomy was assigned to each cluster for which 100% of the assigned taxa were in 100% agreement. Finally, the label “Ambiguous taxa” was removed.

### 2.5. Diversity Analysis

Alpha diversity at the ASV level was assessed using the Shannon index, which reflects both richness and evenness. Beta-diversity analysis was used to evaluate differences in community composition between samples using Bray–Curtis distance metric and visualized by principal coordinate analysis (PCoA). Each dot in the PCoA plot represents one sample. Alpha diversity metrics were calculated with QIIME 2’s “diversity” plugin. Beta diversity analysis was conducted using the phyloseq package (version 1.46.0) in R, within the Bioconductor 3.17 framework. All analyses were performed in a standardized and reproducible computing environment using the Bioconductor Docker image (bioconductor/bioconductor_docker:RELEASE_3_17).

### 2.6. Microbiome Community Typing

Microbial community typing at the genus level was performed using Dirichlet multinomial mixture (DMM) modeling following Hellinger transformation of the abundance matrix. This approach models the gut microbiome composition of each sample as a mixture of different multinomial distributions. The optimal number of clusters (i.e., enterotypes) was determined by fitting the DMM model for a range of possible cluster counts (K), from K = 2 to K = 7. The best-fitting model was selected based on the Laplace approximation to the model evidence. The Laplace approximation serves as a model selection criterion that evaluates how well the model fits the data while penalizing for increased complexity (i.e., a higher number of clusters). The model that minimizes this value is considered optimal because it represents the best balance between goodness-of-fit and parsimony. In our analysis, the Laplace approximation value was lowest when K = 4. Therefore, based on this data-driven criterion, the cohort was stratified into four distinct enterotypes. The analysis was conducted using the DirichletMultinomial package (version 1.46.0) in R (version 4.3.0), within the Bioconductor 3.17 framework [36].

### 2.7. Statistical Analysis

Continuous variables are presented as the mean and standard deviation, whereas categorical variables are presented as frequencies and percentages (n [%]). The relationships between each enterotype and age, physical activity, body composition, and Shannon index were examined using either the Kruskal–Wallis test, or one-way Analysis of Variance (ANOVA), depending on the results of normality and variance homogeneity assessments. Post hoc analyses were performed using Tukey’s test for parametric and the Steel–Dwass test for nonparametric variables. The Shapiro–Wilk test was used to assess normality, and Levene’s test was used to evaluate homogeneity of variances. Fisher’s exact test or Pearson’s chi-square test was applied to compare categorical variables such as J-CHS, KCL, sarcopenia status, and gender across enterotypes. To assess the discriminatory performance for predicting sarcopenia, receiver operating characteristic curves were generated, and predictive accuracy was quantified by the area under the curve (AUC). Multivariable logistic regression, beta diversity analyses, and data visualizations were also performed and DeLong’s test was applied to compare the predictive models. A two-sided *p*-value of <0.05 was considered statistically significant. All statistical analyses were conducted by independent statisticians using JMP Pro 17.0 (SAS Institute Inc., Cary, NC, USA) and R version 4.4.0 (R Foundation for Statistical Computing, Vienna, Austria).

## 3. Results

### 3.1. Subjects’ Characteristics

Study 1, which was based on Cohort 1, initially recruited 434 eligible participants. Among them, 110 participants were excluded without microbiome analysis, and 2 participants were excluded due to missing questionnaire data and physical performance data. A total of 322 participants were analyzed in Study 1, whereas a total of 145 eligible participants were analyzed in Study 2. From Cohort 1, we further excluded 1 participant who did not provide informed consent and 176 participants who had only one microbiome analysis. (Figure 1).

In Cohort 1, the mean age of participants was 76.1 ± 5.5 years, with 96 (29.8%) males and 226 (70.2%) females. In the J-CHS evaluation, 199 (61.8%) subjects were robust, 117 (36.9%) were pre-frailty, and 5 (1.6%) were frailty. In the assessment of sarcopenia, 285 (88.3%) were robust, 32 (9.9%) had sarcopenia, and 5 (1.6%) had severe sarcopenia. The characteristics of Cohort1 subjects are summarized in Table 1, and the characteristics of Cohort 2 subjects are summarized in Appendix A.

The background characteristics of the subjects included in Cohort 1 and Cohort 2 were generally similar. In Cohort 1, the two-step test was unavailable in one participant, respectively; thus, 321 participants were included in the two-step analysis.

### 3.2. Relationship Between Microbiome and Frailty

The Cohort 1 population clustered into four enterotype based on genus-level relative abundance (Figure 2a,b). Enterotypes, defined by dominant gut bacterial genera, are widely utilized for population stratification [16,37]. In this study, four enterotypes were identified as follows:

B1 (ET-B1): High proportion of Bacteroides and Faecalibacterium

B2 (ET-B2): High proportion of Bacteroides and low proportion of Feacalibacterium,

R (ET-R): High proportion of Ruminococcaceae,

P (ET-P): High proportion of Prevotella.

The distribution of enterotypes in Cohort 1 participants was as follows: ET-B1 (n = 79, 24.8%), ET-B2 (n = 82, 25.5%), ET-R (n = 110, 34.2%), and ET-P (n = 51, 15.8%) (Table 1). A significant difference in Shannon index was observed among enterotypes (5.80 ± 0.44 vs. 6.33 ± 0.33, 6.78 ± 0.36, and 6.57 ± 0.39, for B2 vs. B1, R, and P, respectively; *p* < 0.01, one-way ANOVA). Post hoc analysis revealed significant differences between all pairwise comparisons (*p* < 0.01, Tukey’s test). Notably, participants with enterotype ET-B2 had the lowest Shannon index score (Figure 3).

Among the four enterotypes, the percentages of subjects with pre-frailty and frailty according to J-CHS criteria were higher among those with ET-B2 compared with other enterotypes. However, the differences were not significant based on Fisher’s exact test. According to the KCL, the percentage of pre-frailty and frailty subjects with ET-B2 were higher than those for subjects with other enterotypes, but the differences were not significant based on Pearson’s chi-square test. By contrast, among participants in the ET-B2 group, 65 (79.3%) were classified as robust, 15 (18.3%) as having sarcopenia, and 2 (2.4%) as having severe sarcopenia. In the ET-P group, 48 participants (94.1%) were classified as robust, 2 (4.0%) as having sarcopenia, and 1 (2.0%) as having severe sarcopenia. ET-B1 and ET-R groups showed the same tendency regarding sarcopenia. Indeed, a statistically significant difference was observed between sarcopenia and enterotype based on Pearson’s chi-square test (*p* = 0.014) (Table 1).

Sarcopenia is generally evaluated based on gait speed, grip strength and SMI. The gait speed of subjects in the ET-B2 group was 1.34 ± 0.27 m/s, similar to other enterotypes (*p* = 0.25, Welch’s test; Table 1, Figure 4a). By contrast, grip strength differed significantly according to enterotype (24.6 ± 6.87 kg vs. 26.5 ± 7.14 kg, 25.4 ± 7.07 kg, and 28.1 ± 7.63 kg, B2 vs. B1, R, and P, respectively, *p* = 0.034, Kruskal–Wallis test; Table 1). Post hoc analysis indicated that ET-P group had greater grip strength than those in the ET-B2 group (*p* = 0.0498, Figure 4b). No significant differences were observed in two-step value according to enterotypes, as per the Kruskal-Wallis test (Table 1). In terms of body composition, BMI differed significantly according to enterotypes (22.4 ± 3.22 kg/m^2^ vs. 23.5 ± 2.88 kg/m^2^, 22.7 ± 3.01 kg/m^2^, and 23.2 ± 2.87 kg/m^2^, B2 vs. B1, R, and P, respectively, *p* = 0.024, Kruskal–Wallis test; Table 1), and post hoc analysis indicated that ET-B1 group exhibited significantly higher BMI than the ET-B2 group (*p* = 0.044, Figure 4d). A significant difference in SMI was observed according to enterotype (6.20 ± 0.94 kg/m^2^ vs. 6.40 ± 0.88 kg/m^2^, 6.24 ± 0.88 kg/m^2^, and 6.62 ± 0.95 kg/m^2^, B2 vs. B1, R, and P, respectively, *p* = 0.040, Kruskal–Wallis test; Table 1). Although the post hoc analysis did not show any statistically significant pairwise differences, the ET-P group had a higher SMI than the ET-B2 group (*p* = 0.0625, Figure 4e). No significant differences were observed between phase angle and enterotype according to Kruskal-Wallis test (Table 1, Figure 4f).

These findings suggest that particular gut microbiome enterotypes, particularly ET-B2, are associated with sarcopenia. To further investigate clinical factors contributing to sarcopenia, we constructed a multivariable logistic regression model to predict sarcopenia status. The dataset of 322 participants from Study 1 was randomly divided into training and validation subsets in a 7:3 ratio. The training subset included 24 participants with sarcopenia and 201 robust participants, and the validation subset included 13 and 84, respectively. The reduced model, which included age as the sole covariate, yielded an AUC of 0.686. By contrast, the full model incorporating both age and ET-B2 demonstrated significantly improved predictive performance, with an AUC of 0.795 (*p* = 0.045, DeLong’s test; Figure 5).

To examine the stability of enterotypes among participants, the Study 2 population was also clustered into four enterotypes using the same method used in the Study 1 (Appendix A). Approximately 70–80% of participants maintained the same enterotypes (n = 108). However, 37 out 145 participants (25.5%) showed a different enterotype; 12.5% of ET-B1 and 4.3% of ET-R subjects changed to ET-B2 (n = 5 and 2, respectively). 29.4% of ET-B2 subjects changed to ET-B1, ET-P, or ET-R (n = 7 [20.6%], 2 [5.9%], and 1 [2.9%], respectively; Appendix A). Appendix A illustrates the changes in enterotype observed in the Study 2 participants.

## 4. Discussion

In this cross-sectional study, we demonstrated a significant association between the ET-B2 enterotype and sarcopenia among older Japanese adults. Participants with ET-B2 exhibited lower grip strength and SMI compared with ET-P subjects. A multivariable logistic regression model included both age and ET-B2 showed significantly improved predictive accuracy for sarcopenia compared with a model based on age alone.

Sarcopenia is a systemic, age-related muscular disorder characterized by progressive loss of skeletal muscle mass and functional, leading to an increasing risk of falls, fractures, disability, and mortality [38]. Sarcopenia is a multifactorial condition influenced by chronic inflammation [9], immunosenescence [7], anabolic resistance [39,40], and increased oxidative stress [41,42]. Exercise [43], proper nutrition [44], and pharmacological interventions [45] have been studied as potential treatments for sarcopenia; however, no definitive conclusions have been reached regarding effective interventions [46].

In recent years, an association between the gut microbiota and aging has been reported [22], but the relationship between sarcopenia and gut microbiota remains unclear. In the previous study, the features of the gut microbiota in elderly individuals included reduced biodiversity, increased variability in microbiota composition between individuals, and the overgrowth of pathobionts [23]. Jackson et al. reported a negative association between the Frailty Index and gut microbiota alpha diversity in community-dwelling individuals, but participants in that study were not limited to older adults [24]. Claesson et al. found that the microbiota of individual residing in long-stay care facilities is typically significantly less diverse compared with that of a community dweller [25]. However, this study was performed in the absence of established criteria for physical frailty or sarcopenia. Grip strength is a simple and effective metric for measuring muscle strength [26] and used as a diagnostic criterion for sarcopenia [32]. A linear relationship has been reported between baseline grip strength and the incidence of activity of daily life disability [38]. Kang et al. analyzed data from 60 healthy individuals and 27 individuals with sarcopenia or possible sarcopenia and reported that microbial diversity based on the alpha diversity index was reduced in those with sarcopenia or possible sarcopenia [27]. However, their study also had several limitations, such as insufficient sample sizes. Although several studies have reported associations between the gut microbiota and effect of aging, most have limitations that prevent drawing definitive conclusions.

Our study was conducted using gut microbiota data from our previous comprehensive survey based on the JFS. The present study was designed as a relatively large cohort investigation and applied standardized diagnostic criteria for sarcopenia and frailty, utilizing a pattern-based analysis of gut microbiota using enterotypes. Our analysis thus provides a more robust and clinically relevant perspective on the relationship between the gut microbiota and sarcopenia. To the best of our knowledge, this is the first study to demonstrate an association between sarcopenia and gut microbiota enterotypes¸ rather than focusing on individual bacterial taxa.

Metagenomics analysis can provide a comprehensive overview of the gut microbiome, and classification of microbiomes into enterotypes enables pattern-based characterization of the microbial community. Initially, we conducted exploratory analyses at the genus level to examine associations with physical performance, body composition, and frailty scores. However, these analyses did not reveal consistent or statistically meaningful correlations. A comprehensive genus-level analysis to detect robust associations between sarcopenia and gut microbiota would likely require a much larger cohort. Enterotypes are known to be strongly affected by long-term dietary habits [12] and have been associated with various diseases, including obesity, diabetes, and IBD. Vieira-Silva et al. reported the following distribution of four enterotypes among 3515 participants in their study: ET-B1 (36.8%), ET-B2 (13.0%), ET-R (29.4%), and ET-P (20.8%) [20]. Their study included participants from European countries aged 18 to 76 years and excluded those with acute or chronic inflammatory diseases and diabetes. For this reason, we chose to present results primarily at the enterotype level.

Enterotype B is associated with a high-fat, high-protein diet (Western-style meat consumption) [25]. By contrast, P is more common among individuals with high-carbohydrate, fiber-rich diets (agricultural dietary patterns) and among vegetarians [47]. Enterotype P is found predominantly in healthy individuals [48]. The B enterotype can be further divided into B1 and B2 subtypes. The B1 subtype is characterized by a high proportion of Bacteroides/Phocaeicola and Blautia, showing appropriate microbial density and diversity [20]. By contrast, the B2 subtype exhibits features related to dysbiosis, including lower cell counts in fecal samples [49], reduced microbial diversity, increased proportion of Enterobacteriaceae, a lower abundance of beneficial short-chain fatty acid-producing bacteria such as Faecalibacterium, and higher water content [20]. Although Vieira-Silva et al. [20] assessed gene richness rather than alpha diversity, a similar trend was observed in our study, in which the Shannon index of alpha diversity was significantly lower among participants with ET-B2 compared with those with other enterotypes (*p* < 0.01). This suggests that ET-B2 is associated with decreased microbial diversity, potentially reflecting a dysbiotic state. The B2 subtype is associated with systemic inflammation [50], IBD [16], obesity [51], and type 2 diabetes [52]. Further studies are needed, however, to clarify the mechanism by which the B2 subtype contributes to the pathogenesis of these diseases.

The present study has several limitations. First, the composition of the gut microbiota is highly individualized and influenced by a wide range of internal and external factors, including, gender [53,54], environment [50,55], genetic [56], ethnicity [57], dietary habits [12], medication use [58], and age [59]. ET-P was frequently observed in males than in females in our cohort. However, no statically significant association between gender and enterotype was found in the present study based on chi-square tests. To further validate these findings and increase their generalizability, future studies should include more diverse populations across different regions and ethnic backgrounds. Second, the observation that the enterotype changed over time in approximately 25% of participants suggests that the gut microbiota composition is dynamic and potentially modifiable. A longitudinal study of 16 healthy twin pairs reported enterotype changes in 3 cases (18.7%) over approximately 2 years [60]. Additionally, a study involving 41 non-obese participants undergoing a 3-week dietary restriction (40% reduction in daily recommended caloric intake) reported enterotype changes in 5 participants (12.2%) [43]. The higher rate observed in our study may be attributable to differences in classification methods (four enterotype patterns versus two in previous studies) and the older age of our participants, who were more likely to receive medical care and take medications that can influence the gut microbiome. Furthermore, in Japanese older adults, a simulation-based stability analysis revealed a lower resistance index in the lean group than in the normal and obesity groups [61], suggesting that microbiome stability may be reduced in this population. Longitudinal studies are needed to better clarify the temporal relationship between gut microbiota composition and sarcopenia. Third, although the participants were able to engage in cultural circles activities and did not require long-term care—indicating that they were generally healthy and not dependent on medications—we did not confirm the chronic or regular use of proton pump inhibitors, metformin, statins, or probiotics. Therefore, the potential effects of unmeasured or occasional medication use of these or other pharmaceutical agents cannot be entirely excluded. Fourth, the reliability of our sarcopenia prediction model is limited due to the number of participants with sarcopenia (n = 37) and the use of a single-split validation method without cross-validation. Finally, the cross-sectional design precludes causal inferences; therefore, the observed association between ET-B2 and sarcopenia should be interpreted with caution.

## 5. Conclusions

This cross-sectional study demonstrated a significant association between gut microbiota enterotype (ET) and sarcopenia in older Japanese adults. Specifically, participants with the ET-B2 enterotype exhibited lower grip strength and SMI compared with those with the ET-P enterotype. A multivariable logistic regression model that included both age and ET-B2 significantly improved the predictive accuracy for sarcopenia compared with a model based on age alone. These findings highlight the potential of gut microbiota profiles, particularly the ET-B2 enterotype, as biomarkers for age-related muscle decline. Future longitudinal studies are warranted to confirm our results and the relationship between gut microbiota enterotype (ET) and sarcopenia. Interventional studies investigating whether modulation of gut microbiota could contribute to the prevention or treatment of sarcopenia are also needed.

## 6. Patents

Keisuke Hagihara and Aya K. Takeda are planning to submit a patent application related to the results presented in this study.

## Figures and Tables

**Figure 1 nutrients-17-03250-f001:**
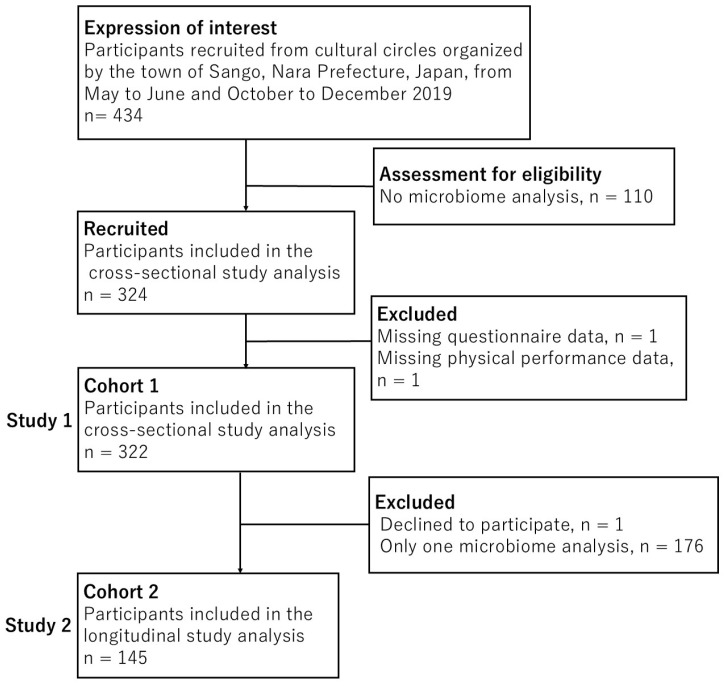
Flowchart of enrollment of subjects for Study 1 (Cohort 1) and Study 2 (Cohort 2).

**Figure 2 nutrients-17-03250-f002:**
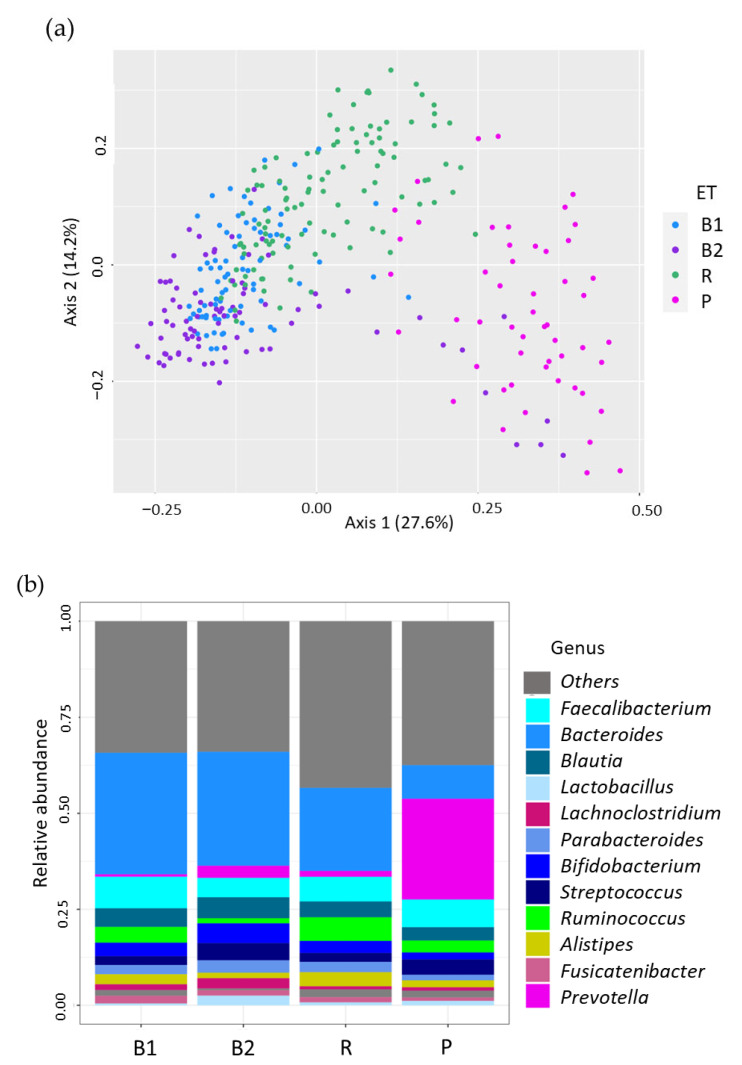
Beta diversity and taxonomic composition by enterotype in Cohort 1. (**a**) Beta-diversity was assessed using principal coordinate analysis based on the Dirichlet multinomial mixtures model, colored by enterotype. (**b**) Taxonomic composition showing the relative abundance of dominant genera by enterotype in Study 1. The twelve most abundant genera are color-coded as indicated in the legend and all other genera are grouped in “Others”.

**Figure 3 nutrients-17-03250-f003:**
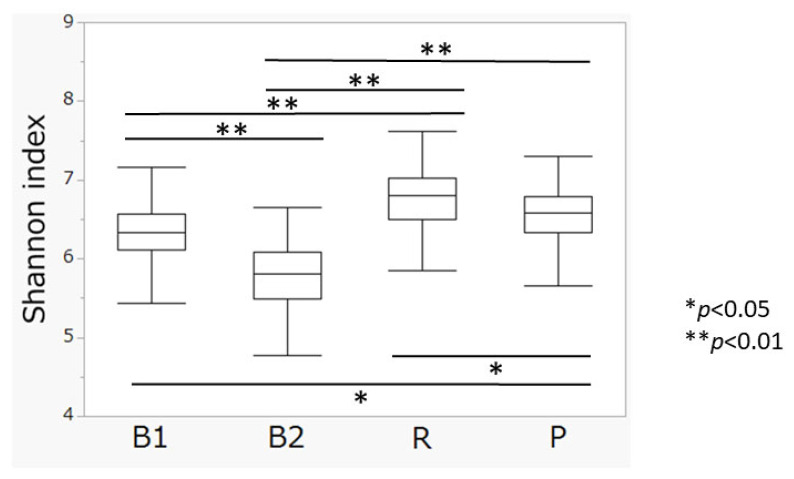
Comparison of Shannon index among enterotypes. Statistical significance was determined using Tukey’s post hoc test (*p* < 0.05; *p* < 0.01).

**Figure 4 nutrients-17-03250-f004:**
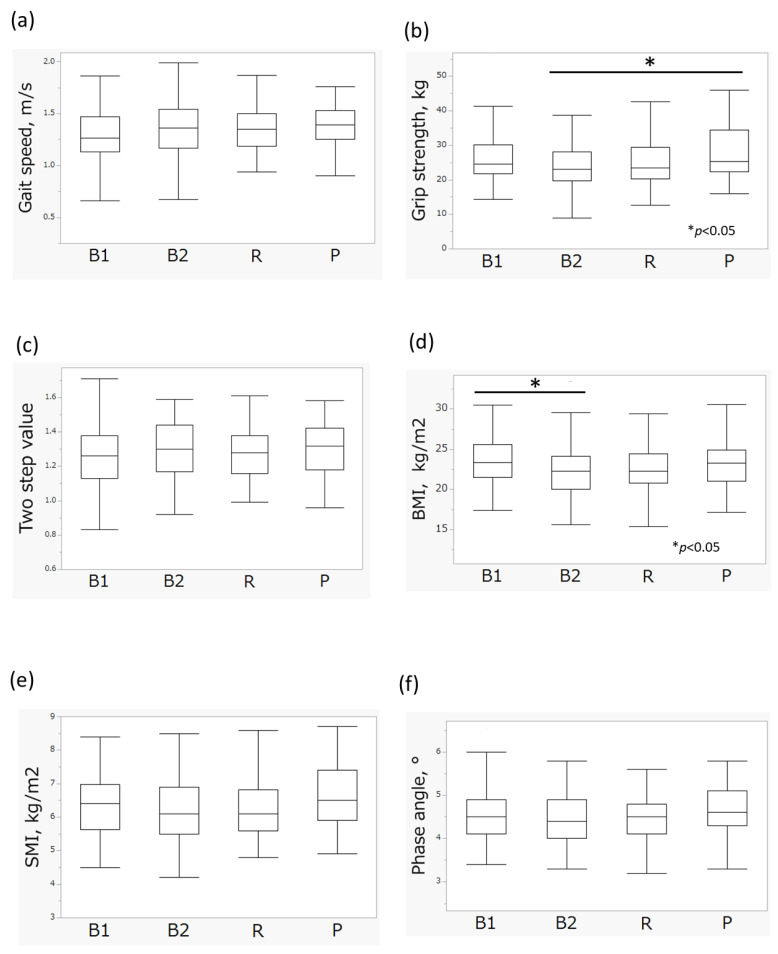
Comparison of physical performance and body composition stratified by enterotype in Cohort 1. (**a**) Gait speed; (**b**) Grip strength; (**c**) Two step value; (**d**) BMI; (**e**) SMI; (**f**) Phase angle. Data are presented as the mean ± SD. *p*-values were calculated using the Steel–Dwass test, as a post hoc analysis, a non-parametric method for multiple comparisons. Statistical significance: *p* < 0.05 for differences among groups. Abbreviation: BMI, body mass index; SMI, skeletal muscle mass index.

**Figure 5 nutrients-17-03250-f005:**
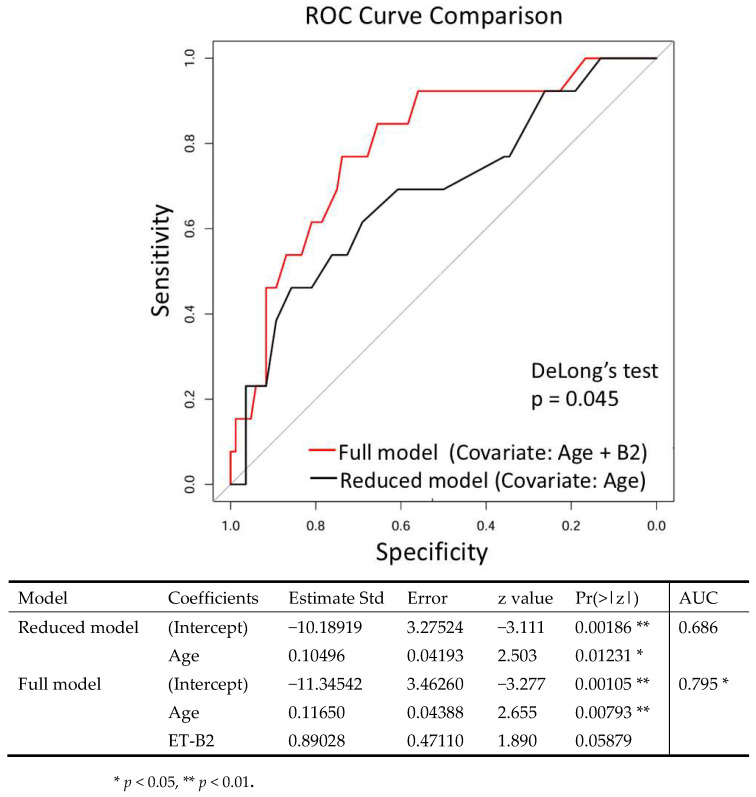
Receiver operating characteristic (ROC) curves comparing the discrimination capacity of the full model and the reduced model for identifying sarcopenia. AUC: 0.686 for the reduced model and 0.795 for the full model. DeLong’s test for AUC comparison: *p* = 0.045. Statistical significance: *p* < 0.05; *p* < 0.01. Abbreviation: AUC, area under the curve.

**Table 1 nutrients-17-03250-t001:** Characteristics of the participants and study outcomes stratified by enterotype in Cohort 1. Abbreviations: J-CHS, Japanese version of Cardiovascular Health Study criteria; KCL, Kihon Checklist; BMI, Body Mass Index; SMI, Skeletal Muscle Mass Index.

Characteristics	Total	Enterotype				
		B1	B2	R	P	*p*-Value
Total, n (%)	322(100)	79 (24.5)	82 (25.5)	110 (34.2)	51 (15.8)	
Male, n (%)	96 (29.8)	23 (24.0)	21 (21.9)	31 (32.3)	21 (21.9)	0.26 ^a^
Female, n (%)	226(70.2)	56 (24.8)	61 (27.0)	79 (35.0)	30 (13.3)	
Age	76.1 ± 5.5	76.1 ± 5.5	74.9 ± 4.8	76.1 ± 5.5	75.7 ± 5.6	0.54 ^b^
J-CHS, n (%)						
Robust	199 (61.8)	50 (63.3)	46 (56.1)	71 (64.6)	32 (62.8)	0.67 ^c^
Pre-frailty	117 (36.3)	26 (32.9)	35 (42.7)	38 (34.6)	18 (35.3)	
Frailty	6 (1.9)	3 (3.8)	1 (1.2)	1 (0.9)	1 (2.0)	
Sarcopenia, n (%)						
Robust	285 (88.5)	74 (93.7)	65 (79.3)	98 (89.1)	48 (94.1)	0.014 ^a^
Sarcopenia	32 (9.9)	3 (3.8)	15 (18.3)	12 (10.9)	2 (4.0)	
Severe sarcopenia	5 (1.6)	2 (2.5)	2 (2.4)	0 (0)	1 (2.0)	
KCL, n (%)						
Robust	164 (50.9)	38 (48.1)	41 (50.0)	58 (52.7)	27 (52.9)	0.61 ^a^
Pre-frailty	124 (38.5)	28 (35.4)	35 (42.7)	42 (38.2)	19 (37.3)	
Frailty	34 (10.6)	13 (16.6)	6 (7.3)	10 (9.1)	5 (9.8)	
Physical activity						
Gait speed(m/s)	1.34 ± 0.24	1.28 ± 0.28	1.34 ± 0.27	1.36 ± 0.20	1.37 ± 0.22	0.25 ^d^
Two-step value	1.28 ± 0.15	1.26 ± 0.17	1.29 ± 0.17	1.28 ± 0.13	1.31 ± 0.14	0.27 ^b^
Grip strength (kg)	25.9 ± 7.19	26.5 ± 7.14	24.6 ± 6.87	25.4 ± 7.07	28.1 ± 7.63	0.034 ^b^
Body composition						
BMI (kg/m^2^)	22.9 ± 3.03	23.5 ± 2.88	22.4 ± 3.22	22.7 ± 3.01	23.2 ± 2.87	0.024 ^b^
SMI (kg/m^2^)	6.33 ± 0.92	6.40 ± 0.88	6.20 ± 0.94	6.24 ± 0.88	6.62 ± 0.95	0.040 ^b^
Phase angle (°)	4.51 ± 0.59	4.55 ± 0.60	4.46 ± 0.57	4.45 ± 0.60	4.66 ± 0.55	0.13 ^b^

^a^: Pearson’s chi-square test, ^b^: Kruskal–Wallis test, ^c^: Fisher’s exact test, ^d^: Welch’s test.

## Data Availability

No data was used for the research described in the article.

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
