# Peer review of "Gut Microbiota Enterotype as a Predictor of Sarcopenia in the Japanese Elderly Population"

_nutrients, 2025, doi:10.3390/nu17203250_

Round 1
Reviewer 1 Report
Comments and Suggestions for Authors
The manuscript investigates the association between gut microbiota enterotypes and sarcopenia in community-dwelling older Japanese adults. This topic is timely and highly relevant, as sarcopenia is an emerging public health concern in aging societies, and the role of gut microbiota in the regulation of muscle mass and function remains insufficiently explored. The authors approach this issue from an interesting perspective by focusing on enterotype-level microbiota patterns rather than on individual bacterial taxa, which adds novelty to the study and may contribute to a better understanding of the gut–muscle axis.
The article follows a classical scientific structure, and the overall presentation is clear. The figures and tables are well designed and easy to interpret, and the statistical methods appear appropriate for the study objectives. The work is generally well organized and informative, with a logical flow from background to discussion. However, several aspects require clarification and improvement.
First, it is not entirely clear why Figure 1 appears in the Materials and Methods section. Based on its content, it seems to represent analytical results rather than a methodological scheme. For clarity and consistency, this figure should be moved to the Results section, where it would naturally fit within the description of enterotype classification and diversity analyses. Its current placement interrupts the methodological narrative and may confuse readers regarding its purpose.
Second, while the authors focused primarily on the enterotype-level analysis, it is not specified whether analyses were also performed at other taxonomic levels such as phylum, family, or genus. It would be valuable to clarify whether similar or stronger associations were observed at these levels, or whether differences were only evident at the enterotype level. The rationale for presenting results exclusively at this taxonomic resolution should be explained. Understanding the reasoning behind this choice would strengthen the methodological transparency and interpretative depth of the study.
Another major concern relates to the rather lenient exclusion criteria. In microbiota research, particularly when fecal samples are analyzed, recent antibiotic use is a key confounding factor that should generally lead to exclusion from the study. The manuscript does not specify whether such information was collected or how it was controlled. Likewise, the use of other medications that may alter gut microbiota composition—such as proton pump inhibitors, metformin, statins, or probiotic supplements—should be clearly addressed. The absence of strict control for these variables could have significantly influenced the observed associations, and this issue requires detailed clarification. At minimum, it should be explicitly discussed as a limitation of the study.
The conclusions section is rather brief and would benefit from expansion. The authors should more clearly emphasize both the potential significance and the limitations of their findings. It is particularly important to acknowledge that the cross-sectional design does not allow for causal inference; therefore, the observed association between the ET-B2 enterotype and sarcopenia should be interpreted with caution. A stronger discussion of the implications for future longitudinal and interventional studies would make the manuscript more comprehensive and forward-looking.
Finally, the manuscript contains several minor editorial and stylistic issues. There are typographical and grammatical errors.
I redcommend major revision.
Author Response
We sincerely appreciate your insightful comments and constructive suggestions. We have carefully addressed each point raised as follows:
Comments 1: First, it is not entirely clear why Figure 1 appears in the Materials and Methods section. Based on its content, it seems to represent analytical results rather than a methodological scheme. For clarity and consistency, this figure should be moved to the Results section, where it would naturally fit within the description of enterotype classification and diversity analyses. Its current placement interrupts the methodological narrative and may confuse readers regarding its purpose.
Response 1: Thank you for your comments. We changed Figure 1 to Figure 2b and replaced from the Materials and Methods section to the Results section at page 7.
Comments 2: Second, while the authors focused primarily on the enterotype-level analysis, it is not specified whether analyses were also performed at other taxonomic levels such as phylum, family, or genus. It would be valuable to clarify whether similar or stronger associations were observed at these levels, or whether differences were only evident at the enterotype level. The rationale for presenting results exclusively at this taxonomic resolution should be explained. Understanding the reasoning behind this choice would strengthen the methodological transparency and interpretative depth of the study.
Response 2: Initially, we conducted exploratory analyses at the genus level, assessing associations with physical performance, body composition, and frailty scales. However, these analyses did not reveal consistent or statistically meaningful correlations. A comprehensive genus-level analysis to detect robust associations between sarcopenia and gut microbiota would likely require a much larger cohort. Enterotypes are known to be strongly affected by long-term dietary habits and have been associated with various diseases. For this reason, we chose to present results primarily at the enterotype level. We have clarified this rationale in the Discussion sections in Line 360-365.
Comments 3: Another major concern relates to the rather lenient exclusion criteria. In microbiota research, particularly when fecal samples are analyzed, recent antibiotic use is a key confounding factor that should generally lead to exclusion from the study. The manuscript does not specify whether such information was collected or how it was controlled. Likewise, the use of other medications that may alter gut microbiota composition—such as proton pump inhibitors, metformin, statins, or probiotic supplements—should be clearly addressed. The absence of strict control for these variables could have significantly influenced the observed associations, and this issue requires detailed clarification. At minimum, it should be explicitly discussed as a limitation of the study.
Response 3: Thank you for your comments. In the previous cohort study, the participants over 65 years or older were able to participate cultural circles activities. The exclusion criterion was certified as needing long-term care. Thus, participants basically did not need medications., such as chronic or regular use of proton pump inhibitors, metformin, statins, or probiotics. We added the explanations in the limitation section (page 13, Line 410-414).
Comments 4: The conclusions section is rather brief and would benefit from expansion. The authors should more clearly emphasize both the potential significance and the limitations of their findings. It is particularly important to acknowledge that the cross-sectional design does not allow for causal inference; therefore, the observed association between the ET-B2 enterotype and sarcopenia should be interpreted with caution. A stronger discussion of the implications for future longitudinal and interventional studies would make the manuscript more comprehensive and forward-looking
Response 4: The conclusions have been expanded to emphasize both the potential significance and the limitations of our findings. We explicitly note that the cross-sectional design precludes causal inferences, and that the association between ET-B2 and sarcopenia should be interpreted with caution. Implications for future longitudinal and interventional studies have also been added in(Line 421-431).
Comments 5: Finally, the manuscript contains several minor editorial and stylistic issues. There are typographical and grammatical errors.
Response 5: This manuscript has been reviewed and edited by a native English speaker.

Reviewer 2 Report
Comments and Suggestions for Authors
This cross-sectional study of 322 older Japanese adults found that the gut microbiota enterotype ET-B2 was significantly associated with sarcopenia, characterized by lower microbial diversity, reduced grip strength, and a trend toward lower skeletal muscle mass. The paper presents its findings in a clear and coherent manner, but I have concerns about the enterotype classification. Here are my comments:
- Line 48 “dysbiosis”, to clarify that the discussion is focused on the microbial community in the gut, please use "gut microbiome dysbiosis" where appropriate.
- The introduction would benefit from a concluding paragraph that clearly states the specific objective and approach of this study and briefly highlights its potential significance in the field.
- Section 2.6, please specify the criteria used for enterotype clustering.
- Please clarify what "Study 1" and "Study 2" refer to. If they are different patient groups within this project, would "Cohort 1" and "Cohort 2" be more appropriate and less ambiguous terms?
- Lines 278-285, to strengthen the validity and translational potential of the model developed in Study 1, I strongly recommend using it to predict the outcomes in the independent Study 2.
- The model parameters presented in Figure 6 should be organized into a clear table.
- The conclusion section is overly brief.
Author Response
Overview comments
The paper presents its findings in a clear and coherent manner, but I have concerns about the enterotype classification.
Response
We appreciate your understanding and positive comments on our study. Considering your concern about the gut microbiota enterotype classification, we improved our draft and added the sentence about our method for the gut microbiota enterotype determination, as detailed below.
Comments 1: Line 48 “dysbiosis”, to clarify that the discussion is focused on the microbial community in the gut, please use "gut microbiome dysbiosis" where appropriate.
Response 1: We appreciate the reviewer’s helpful comment. In accordance with the suggestion, we have replaced “dysbiosis” with “gut microbiome dysbiosis” in line 48 to clarify that the discussion focuses on the gut microbial community.
Comments 2: The introduction would benefit from a concluding paragraph that clearly states the specific objective and approach of this study and briefly highlights its potential significance in the field.
Response 2: Thank you for your comments. Based on yours and other reviewer’s comments, we have revised the Introduction session page2, Line 58-71.
Comments 3: Section 2.6, please specify the criteria used for enterotype clustering.
Response 3: Thank you for your helpful comments. The enterotype clustering was performed using the Dirichlet multinomial mixtures (DMM) method, a probabilistic clustering method specifically suited for microbial abundance data. This approach models the gut microbiome composition of each sample as a mixture of different multinomial distributions. The optimal number of clusters (i.e., enterotypes) was determined by fitting the DMM model for a range of possible cluster counts (K), from K=2 to K=7. The best-fitting model was selected based on the Laplace approximation to the model evidence. The Laplace approximation serves as a model selection criterion that evaluates how well the model fits the data while penalizing for increased complexity (i.e., a higher number of clusters). The model that minimizes this value is considered optimal because it represents the best balance between goodness-of-fit and parsimony. In our analysis, the Laplace approximation value was lowest when K=4 (data not shown). Therefore, based on this data-driven criterion, the cohort was stratified into four distinct enterotypes. This clarification has been added to Section 2.6. Microbiome community typing(Line173-185).
Comments 4: Please clarify what "Study 1" and "Study 2" refer to. If they are different patient groups within this project, would "Cohort 1" and "Cohort 2" be more appropriate and less ambiguous terms?
Response 4: Thank you for your helpful comments regarding the description of the study population. To improve clarity, we have revised the Methods section to specify that Study 1 corresponds to Cohort 1 (cross-sectional study) and Study 2 corresponds to Cohort 2 (longitudinal study) in the Results session and Figure 1.
Comments 5: Lines 278-285, to strengthen the validity and translational potential of the model developed in Study 1, I strongly recommend using it to predict the outcomes in the independent Study 2.
Response 5: Thank you for your suggestion regarding the use of study 2 for additional model validation. While we understand the importance of external validation to strengthen the model’s generalizability, study 2 is not an independent dataset but rather a subset of Study 1, consisting of participants with repeated measurements across multiple time points. Therefore, there is a complete inclusion relationship between the two datasets, and using study 2 as a validation cohort would not ensure statistical independence, potentially leading to overfitting or inflated performance estimates.
Comments 6: The model parameters presented in Figure 6 should be organized into a clear table.
Response 6: Thank you for your helpful comments. We have organized the model parameters into a clear table and revised the caption of Figure 5 accordingly.
|
Model |
Coefficients |
Estimate Std |
Error |
z value |
Pr(>|z|) |
AUC |
|
Reduced model |
(Intercept) |
-10.18919 |
3.27524 |
-3.111 |
0.00186** |
0.686 |
|
|
Age |
0.10496 |
0.04193 |
2.503 |
0.01231* |
|
|
Full model |
(Intercept) |
-11.34542 |
3.46260 |
-3.277 |
0.00105** |
0.795* |
|
|
Age |
0.11650 |
0.04388 |
2.655 |
0.00793** |
|
|
|
ET-B2 |
0.89028 |
0.47110 |
1.890 |
0.05879 |
|
Comments 7: The conclusion section is overly brief.
Response 7: Thank you for your comments. We have revised the Conclusion session in page 13, Line 421-431.

Reviewer 3 Report
Comments and Suggestions for Authors
Comments to the Authors of manuscript number nutrients-3920505 entitled "Gut Microbiota Enterotype as a Predictor of Sarcopenia in the Japanese Elderly Population"
- Lines 25–27 (Abstract) "Trend toward lower SMI … p = 0.0625" – The authors describe the result as a "trend." Values ​​p > 0.05 should not be interpreted as a trend, but as a lack of significance.
- Lines 56–57 (Introduction) "JFS exhibited low correlation with measures of physical performance such as grip strength and SMI." Since JFS does not correlate with muscle parameters, its use as a starting point for studying sarcopenia is questionable and requires better justification.
- Lines 105–106 "Sarcopenia is similar to frailty" – This simplification is factually incorrect. Sarcopenia is a distinct syndrome, partially overlapping with frailty, but not directly "similar."
- Line 176 "Bateroides" – Bacteroides.
- Lines 201–206 (Results – Subjects) The numbers are inconsistent: "434 eligible participants," "110 excluded," "2 excluded," leaving 322. Then "we excluded 1 participant … and 32 ..." – then the result is not 322, but 289. There is also ambiguity in Figure 2.
- Lines 249–251 "A statistically significant difference was observed between sarcopenia and enterotype based on Fisher's exact test and Pearson's chi-square test." The authors should clearly state which test yielded a result of p=0.014.
- Line 257–258 (Figure 5b) "p = 0.0498" – a very borderline value; avoid labeling this as "significant" without correcting for multiple comparisons.
- Line 263–267 (SMI) The authors write about "tendency" at p = 0.0625 – a similar issue to the one in the Abstract.
- Line 273–277 (ROC curves) Randomly splitting 322 samples into training/test without cross-validation with such a small number of sarcopenia cases (n = 37) is methodologically questionable. It should be clearly stated that the model has limited reliability.
- Line 314–316 (Discussion) "...However, this study was performed in the absence of..." – double period and illogical merging of sentences.
- Lines 363–364 (Discussion) "ET-P is more common among males than in females." Sex differences are more complex and do not necessarily indicate a P predominance in males.
- Lines 369–374 (Discussion) The authors compare the enterotype variability to other studies (18.7%, 12.2%), but in their case, the change occurred in 25.5%. They should clearly discuss their higher frequency and possible reasons (classification method, observation time).
- Figure 1: No percentage scale on the Y-axis; difficult to compare the proportion of dominant types. Colors not described in the legend.
- Figure 2 (flowchart): As mentioned, inconsistencies in the numbers; the diagram does not match the description in the text.
- Figure 3 (PCoA): overlapping points, no percentage variance on the axes.
- Figure 4 (Shannon index): the test used for comparisons is missing in the legend.
- Figure 5 (performance and composition): the caption indicates that the test was the Steel–Dwass test, but the Kruskal–Wallis and Tukey test were previously used in the text.
- Figure 6 (ROC curves): the number of sarcopenia cases in the validation set is missing.
19.sample size inconsistencies (lines 201–206, Figure 2),
20 incorrect inferences from p > 0.05 (lines 25–27, 263–267),
21 inconsistencies in statistical tests and figure legends.
Author Response
Comments 1: Lines 25–27 (Abstract) "Trend toward lower SMI … p = 0.0625" – The authors describe the result as a "trend." Values ​​p > 0.05 should not be interpreted as a trend, but as a lack of significance.
Response 1: Thank you for your helpful comments. We have revised the description to remove the expression “trend toward lower SMI” and now state that the difference in SMI compared with ET-P individuals did not reach statistical significance (p = 0.0625) in Line 25-28.
Comments 2: Lines 56–57 (Introduction) "JFS exhibited low correlation with measures of physical performance such as grip strength and SMI." Since JFS does not correlate with muscle parameters, its use as a starting point for studying sarcopenia is questionable and requires better justification.
Response 2: Thank you for your comments. Based on yours and other reviewer’s comments, we have revised the Introduction session page2, Line 58-71.
Comments 3: Lines 105–106 "Sarcopenia is similar to frailty" – This simplification is factually incorrect. Sarcopenia is a distinct syndrome, partially overlapping with frailty, but not directly "similar."
Response 3: Thank you for your comments. We have revised the sentence to clarify that sarcopenia is partially overlapping with frailty in Line 114-117.
Comments 4: Line 176 "Bateroides" – Bacteroides.
Response 4: Thank you for your comments. We have revised the sentence “Bacteroides” in Line 240.
Comments 5: Lines 201–206 (Results – Subjects) The numbers are inconsistent: "434 eligible participants," "110 excluded," "2 excluded," leaving 322. Then "we excluded 1 participant … and 32 ..." – then the result is not 322, but 289. There is also ambiguity in Figure 2.
Response 5: Thank you for your comments. We apologize for the simple mistake. We have revised the Figure 1 and Line 213-215.
Comments 6: Lines 249–251 "A statistically significant difference was observed between sarcopenia and enterotype based on Fisher's exact test and Pearson's chi-square test." The authors should clearly state which test yielded a result of p=0.014.
Response 6: Thank you for your comments. We have revised Lines 269–271 to clearly state that the result of p=0.014 was yielded by Pearson's chi-square test.
Comments 7: Line 257–258 (Figure 5b) "p = 0.0498" – a very borderline value; avoid labeling this as "significant" without correcting for multiple comparisons.
Response 7: Thank you for your helpful comments. We have deleted ‘significantly’ and revise the sentence in Line 276-278 accordingly.
Comments 8: Line 263–267 (SMI) The authors write about "tendency" at p = 0.0625 – a similar issue to the one in the Abstract.
Response 8: Thank you for your comments. We have deleted “tend to” and revised the sentence in Line 286-289 to present the finding more objectively.
Comments 9: Line 273–277 (ROC curves) Randomly splitting 322 samples into training/test without cross-validation with such a small number of sarcopenia cases (n = 37) is methodologically questionable. It should be clearly stated that the model has limited reliability.
Response 9: Thank you for your comments. We agree with the reviewer that the model reliability is limited due to the number of sarcopenia cases and the use of a single-split validation. We have revised the Discussion section accordingly and added this point to the limitations (Line 415-417).
Comments 10: Line 314–316 (Discussion) "...However, this study was performed in the absence of..." – double period and illogical merging of sentences.
Response 10: Thank you for pointing this out. We have corrected the double period and revised the sentence in Line 339-341 for clarity.
Comments 11: Lines 363–364 (Discussion) "ET-P is more common among males than in females." Sex differences are more complex and do not necessarily indicate a P predominance in males.
Response 11: Thank you for your comments. We have deleted and revised the sentence in Line392-393.
Comments 12: Lines 369–374 (Discussion) The authors compare the enterotype variability to other studies (18.7%, 12.2%), but in their case, the change occurred in 25.5%. They should clearly discuss their higher frequency and possible reasons (classification method, observation time).
Response 12: Thank you for raising this point. The higher frequency of enterotype change observed in our study (25.5%) may be explained by several factors. First, ET-B1 and ET-B2 are similar composition. Other studies were classified participants into two enterotype patterns. As a result, other studies cannot capture these changes. Second, our participants were older than in other studies. Older adults are more likely to receive medical care and take medications, such as antibiotics, which could influence the gut microbiome enterotype and lead to enterotype change. Furthermore, in Japanese older adults, a simulation-based stability analysis revealed a lower resistance index in the lean group than those in the normal and obesity groups [61], suggesting that microbiome stability may be reduced in this population. We have added this explanation to the Discussion session (Line402-408).
Comments 13: Figure 1: No percentage scale on the Y-axis; difficult to compare the proportion of dominant types. Colors not described in the legend.
Response 13: Thank you for your comments. We have replaced the Figure from the Materials and Method section to the Result section as Figure 2b and clarified the color representations in the legend. The Y-axis represents Relative abundance, which is displayed as a proportion from 0 to 1 and is therefore not a percentage scale.
Comments 14: Figure 2 (flowchart): As mentioned, inconsistencies in the numbers; the diagram does not match the description in the text.
Response 14: Thank you for your comments. We have revised Figure 1 same as Response 5.
Comments 15: Figure 3 (PCoA): overlapping points, no percentage variance on the axes.
Response 15: Thank you for your comments. The percentage of variance explained (contribution rate) is clearly indicated on the PCoA axes: PC1 (27.6%) and PC2 (14.2%). The overlap of points you observed is a typical feature of PCoA plots for large datasets like ours (N > 300). It reflects the continuous nature of the microbial community data, where samples do not form perfectly discrete clusters but rather exist along a spectrum.
Comments 16: Figure 3 (Shannon index): the test used for comparisons is missing in the legend.
Response 16: Thank you for your comments. We have revised the legend in Figure 3 to indicate that comparisons were performed using Tukey’s post hoc test.
Comments 17: Figure 5 (performance and composition): the caption indicates that the test was the Steel–Dwass test, but the Kruskal–Wallis and Tukey test were previously used in the text.
Response 17: Thank you for your comments. As mentioned in Material and method section, we used the one-way ANOVA followed by Tukey’s test for post hoc comparison of Shannon index, which is normally distributed. We used the Kruskal-Wallis test for overall comparison, followed by the Steel-Dwass test for post hoc pairwise comparisons of physical parameters, which are not normally distributed. We have revised the sentence in Line 309-310
Comments 18: Figure 6 (ROC curves): the number of sarcopenia cases in the validation set is missing.
Response 18: Thank you for your comments. We have added the numbers of sarcopenia and robust participants in the training (24 and 201) and validation subset (13 and 84) to the manuscript in Line 294-295.
Comments 19: sample size inconsistencies (lines 201–206, Figure 2),
Response 19: Thank you for your comments. We revised Figure 1(Flow chart) and sample size is same as described comment 5.
Comments 20: incorrect inferences from p > 0.05 (lines 25–27, 263–267),
Response 20: Thank you for your comments. We have revised the sentences as described in Comment1,8.
Comments 21: inconsistencies in statistical tests and figure legends.
Response 21: Thank you for your comments. We have revised figure legends of Figure 3, 4, and Figure 5,

Round 2
Reviewer 2 Report
Comments and Suggestions for Authors
I thank the authors for addressing all my comments. I have no further concerns.
Reviewer 3 Report
Comments and Suggestions for Authors
thank you for the correction